# Effects of Induced Exosomes from Endometrial Cancer Cells on Tumor Activity in the Presence of *Aurea helianthus* Extract

**DOI:** 10.3390/molecules26082207

**Published:** 2021-04-12

**Authors:** Yoonjin Park, Kyunghwa Lee, Suhng Wook Kim, Min Woo Lee, Boyong Kim, Seung Gwan Lee

**Affiliations:** 1Department of Clinical Laboratory Sciences, College of Health Science, Korea University, Seoul 02841, Korea; pyoonjin@naver.com (Y.P.); swkimkorea@korea.ac.kr (S.W.K.); leemw@korea.ac.kr (M.W.L.); 2Transdisciplinary Major in Learning Health Systems, Graduate School, Korea University, Seoul 02841, Korea; 3Life Together, 13 Gongdan-ro, Chuncheon-si 24232, Gangwon, Korea; 4Mitosbio, 13 Gongdan-ro, Chuncheon-si 24232, Gangwon, Korea; kh_lee@lifetogether.co.kr

**Keywords:** *Aurea helianthus*, endometrial cancer, exosome, migration, microRNA, uterus

## Abstract

Endometrial cancer (EC) cells metastasize to various regions, including the ovaries, fallopian tubes, cervix, blood, liver, bone, and brain. Various carcinogens are known to cause EC. Exosomes are released from several types of cells and contain various cellular components. In this study, flow cytometry and quantitative PCR were used to evaluate marker levels, cell migration, cell invasion, and mitochondrial membrane potential, and cellular senescence tests were used to estimate cancer activity. The microRNAs were profiled using next-generation sequencing. Although tocopherol-α and rutin content in *Aurea helianthus* is high, *A. helianthus* extract was more useful in modulating tumor activity compared to the two aforementioned substances. Notably, we established that the extract induced bioactive exosomes in EC cells, and profiling of miRNAs in the extract-inducing exosomes (EIE) indicated their potency to be developed as a biological drug. The extract and EIE contributed to the following five biological process categories for EC cells: (1) cell migration and invasion suppression, (2) cellular senescence activation by attenuating mitochondrial membrane potential and enhancing autophagy, (3) reproductive cancer activity attenuation, (4) drug susceptibility activation, and (5) EIE containing miRNAs associated with decreasing inflammation.

## 1. Introduction

Endometrial cancer (EC) is classified into two types: type I, the most common type, and type II [1]. Cancer cells metastasize to various regions, including the ovaries, fallopian tubes, cervix, blood, liver, bone, and brain [2,3]. EC is caused by multiple carcinogens, including chemicals, radiation, and biological reasons, such as hormonal imbalance and human papillomavirus (HPV) infection [4]. Recently, the incidence of EC in females has increased steadily, and the incidence of type I EC has risen significantly globally [1,5]. The causes of EC are obesity, excessive estrogen, high blood pressure, and diabetes mellitus [6].

The typical characteristics of activated cancer cells include cell death evasion, healthy metabolism, unlimited replication, tissue invasion, and metastasis [7]. Several studies have reported that EC and cervical cancer cells express specific markers, including squamous cell carcinoma antigen (SCC), urinary gonadotropin fragment (UGF), Muc16 (CA-125; cancer antigen 125), and cytokeratin 8 (CK8) [8,9,10,11]. Metastasis is a significant characteristic of malignant tumors, and preventing metastasis is an essential therapeutic target [12]. During metastasis activation, cervical cancer cells express several genes and proteins, including the homeobox (*HOX*) gene, PI3K/AKT/mTOR, epidermal growth factor receptor (EGFR), platelet-derived growth factor receptor (PDGFR), vascular endothelial growth factor (VEGF), and vimentin [13,14,15,16,17].

Exosomes are extracellular vesicles released from several types of cells [18]. In particular, cancer cells release more exosomes with different compositions than normal cells. Tumor-derived exosomes contribute to proliferation, inflammation, drug resistance, metastasis, tumorigenesis, and immune response and possess bioactive compounds, including proteins, single-stranded and genomic DNA, retrotransposon elements, messenger RNA, long non-coding RNA, and microRNAs (miRNAs) [18,19]. Exosomal miRNAs modulate the expression of oncogenes and activate suppressors in tumor cells [18]. Moreover, their alteration functions as a critical modulator of carcinogenesis, and under a functional phytoextract, miRNAs in tumor cells are expected to be improved to alter tumor activity.

*Aurea helianthus*, belonging to the family *Malvaceae*, has various functions, including inflammation, fever, and tumor suppression; oxidation and melanogenesis; detoxification; lipid metabolism modulation; and immune regulation [20]. Notably, the leaves have a high tocopherol-α and rutin content, in addition to iron, vitamin A, and vitamin C [20]. Tocopherol-α, a soluble phenolic compound, is a type of vitamin E and potent antioxidant that prevents cancer activity [21]. Additionally, rutin, known to inhibit EGFR kinase, protects DNA structure and induces apoptosis of cancer cells through endoplasmic reticulum stress [22]. According to previous reports [20], *A. helianthus* extract (AH) protects human vaginal cells from oxidative and fungal stress and strongly suppresses CK8, Muc-16, and vimentin expression. Furthermore, it modulates monocyte differentiation and activates HPV peptide phagocytosis [20].

Although no prior study has focused on the functions of phytoextract-inducing exosomes to date, the present study documented the potency of specific miRNAs in AH-induced exosomes (EIE), in addition to the functions of the extracts, including EC marker expression downregulation, and EC cell invasion and migration suppression. The functional miRNAs in this study play a key role in the development of a biological drug to protect against or heal gynecological cancers through in vivo experiments. In particular, the results of this study suggest that EIE robustly protects against and attenuates EC, and miRNAs in EIE may be utilized to develop liposomes for use as a biological drug.

## 2. Results

This study targeted to categorize the roles of AH and EIE in modulating tumor activity into five categories (Table 1). Furthermore, based on the miRNA profiling in EIE, miRNAs significantly involved in each category were identified in EC cells (Graphic abstract).

### 2.1. Expression of EC Cell Markers in the Presence of Bioactive Substances

The extracts effectively downregulated SCC and UGF genes in EC cells (Figure 1a). In contrast to 25 µg/mL rutin (R25), 5 µg/mL tocopherol-α (TP5) and 100, 500, and 1000 µg/mL AH (AH100, AH500, AH1000, respectively) effectively downregulated SCC (Figure 1a). In the presence of AH1000, the SCC level was approximately two times lower than that in the presence of TP5 (Figure 1a). Moreover, AH also downregulated the UGF gene, which was strongly downregulated in EC cells in the presence of AH1000 (Figure 1a).

Compared with the expression of all markers in control cells, the expression of each marker decreased in EC cells in the presence of all bioactive substances (Figure 1b). The number of marker-positive CK8^+^ cells in the presence of R25 and AH was two and four times lower than that of marker-positive control cells, respectively (Figure 1b). Although TP5 and R25 did not decrease the number of marker-positive Muc-16^+^ cells significantly, it was significantly decreased in the presence of AH1000 (Figure 1b). However, R25 and TP5 decreased the number of marker-positive vimentin^+^ cells slightly, while AH1000 decreased the number of marker-positive vimentin^+^ cells by a factor of 5 compared to the number of marker-positive control cells (Figure 1b).

### 2.2. Inhibition of Tumor Activity by Bioactive Substances

AH and TP5 significantly inhibited EC migration after 36 h, in contrast to the enhanced EC migration in the control (Figure 2a). EC migration inhibition in the presence of AH1000 was seven times higher than that in the control, while that in the presence of AH100 and AH500 was about two times higher than that in the presence of TP5. Likewise, vimentin expression and EC migration inhibition in the presence of R25 were not different from those in the control cells (Figure 1b and Figure 2a). Interestingly, only AH1000 strongly suppressed the EC cell invasive activity at 12 h, which was 1.56 times lower than that in the control cells (Figure 2b). These substances also affected mitochondrial membrane potential (MMP) and cellular senescence. TP5 did not affect MMP, although MMP in the presence of R25 was 1.5 times lower than that in the control, while that in the presence of AH500 and AH1000 was significantly lower than that in the presence of R25 (Figure 2c). The cancer cells were senescenced by all substances. Notably, the senescence by AH1000 was approximately 4.16 and 2.1 times higher than that by TP5 and R25, respectively (Figure 2d).

### 2.3. Expression of Cancerous Markers by the Induced Exosomes

The induced CD63^+^-exosomes were isolated from EC cells in the presence of the three bioactive substances (Figure 3a). The EC cells were exposed to exosomes that downregulated the expression of markers, including SCC, UGF, and IL2-receptor (Figure 3b). Additionally, EIE downregulated the expression of drug resistance-associated genes, including *NF-κB (P50, P52), mTORC2*, and *ABCB1* (Figure 3c). Furthermore, the EIE and TP downregulated Muc-16^+^vimentin^+^ cell count by 0.37 and 0.45 times, respectively (Figure 3d).

### 2.4. Effects of Induced Exosomes

For profiling the miRNAs in the EIE (Table 1), EC cells were exposed to EIE (Figure 4a). The RNA levels of the evaluating genes (Table 1) were modulated by targeting the miRNA in the five biochemical categories (Figure 4 and Table 1). Interestingly, the RNA level results for the evaluated genes corresponded with the profiling (Figure 4a and Table 1). Notably, *PTGS*, *ESR*, and *HSD17B12* levels decreased by more than four times compared with the control EC cells (Figure 4a). Additionally, this study documented the effects of EIE on modulating the microenvironment in ovarian cancer (Figure 4b). Interestingly, the levels of estrogen and androgen receptors were significantly downregulated in EIE-treated ovarian cancer cells (Figure 4b).

### 2.5. Profiling of miRNAs in the Induced Exosomes

The clustering heatmap displayed the patterns of significantly altered miRNAs based on the analytical results for miRNAs in the exosomes induced by the three substances (Figure 5a). Unlike the heatmap pattern in rutin-induced exosomes, the heatmap in AH1000-induced exosomes was significantly altered (Figure 5a). Moreover, the miRNA distribution on the scatter plot of AH1000- and control-induced exosomes was the most different, wherein 112 and 102 miRNAs in EIE were upregulated and downregulated, respectively, in EC cells (Figure 5b,c). Although some miRNAs involved in various categories were unaltered in rutin-induced exosomes, the miRNAs in the AH1000- and TP5-induced exosomes were downregulated (Figure 5d).

Several miRNAs associated with the five biochemical categories were significantly altered in the EIE (Figure 5e and Table 1). The results indicate that EIE modulated specific bioactivities, including autophagy, drug transport and carbolic processes, cell migration, endocrine metabolism, cellular respiration, and immune response. Notably, the levels of hsa-miR-423-5p and hsa-miR-1908-5p were 100 times lower than those in the control (Figure 5e, Table 1). Moreover, the miRNAs involved in several categories were significantly altered (Figure 5e and Table 1).

## 3. Discussion

EC is the most common gynecological malignancy and malignant tumors are associated with obesity [23]. Excess estrogen causes EC, insulin resistance, and obesity-driven inflammation, and EC incidence rate has been increasing with increase in obesity prevalence [24]. The novel aim and results of this study will be crucial for the development of effective clinical materials using a functional phytoextract with minimal side effects and maximal effects. This study documented the suppression of cancer activity by three bioactive substances and compared their relative efficacy and functional alteration of the induced exosomes in EC cells. Furthermore, this study suggests functional miRNA candidates for the development of pharmaceutical materials for gynecological cancers. In further studies, we have investigated the synthesis of liposomes containing functional miRNAs based on the candidates. Moreover, the functions of the synthesized liposomes have been estimated in ovo and in vivo to develop pharmaceutical materials.

Compared to tocopherol-α and rutin, the AH had two significant characteristics. It suppressed various cancer activities, including cancer metabolism, migration, and invasiveness and induced significant miRNA alteration in the EIE (Graphic abstract). Unlike tocopherol-α and rutin, AH, particularly AH1000, significantly downregulated SCC and UGF (Figure 1). SCC binds to carbonyl reductase, which inhibits malignant behavior and TGF-β signaling in uterine cancer cells [25].

Additionally, SCC inhibits cellular apoptosis by irradiation, natural killer cells, anticancer drugs, and irradiation, and promotes migration and invasion by decreasing E-cadherin [26,27,28]. UGF, the β-subunit of human chorionic gonadotropin (hCGβ), also suppresses cellular apoptosis [29] and promotes migration, invasion, malignant transformation, and drug resistance [30,31,32]. These results suggest that AH1000 effectively suppressed the malignant behavior of EC cells, and is a good material for preventing malignant behavior and cancerous transformation of normal endometrial cells. These predicted functions were shown by the results in biological categories, including migration, invasion, and expression of malignant markers, cellular senescence, and MMP (Figure 2).

The extract and EIE modulate tumor activity around EC cells through their effects on five biological categories. In EC bearing and genetic models [33,34,35], progesterone and estrogen receptors, mTOR signaling are positive in mice. In addition, invasive endometrial tumor with PTEN (phosphatase and tensin homolog) /LKB1(liver kinase B1)-deficiency displays dysregulated Lkb1/Ampk and phosphatidylinositol 3-kinase (PI3K)/Akt signaling with intensive mTOR signaling [34]. Based on these studies, the following five characteristics of EIE suggest a potency to protect against and heal gynecological cancers.

First, tocopherol-α and the extracts significantly inhibited the malignant behavior of EC cells (Figure 2a,b). In particular, the AH1000-induced exosomes contained the migration inhibitory miRNAs, such as hsa-miR-1908-5p and hsa-miR-20b-5p. Interestingly, their levels were 100 times and 8.2 times higher than those in the control (Table 1 and Figure 5). The genes evaluated for hsa-miR-1908-5p and hsa-miR-20b-5p were prostaglandin endoperoxide synthase 2 (*PTGS2*) and *VEGFA*, respectively. The PTGS2 protein level is modulated by beta-2 adrenergic receptor (ADRB2) receptor signaling, and PTGS2 silencing suppresses migration and invasion of ovarian cancer cells [36]. VEGF protein has critical roles, including migration, invasion, angiogenesis, endothelial cell proliferation, invasion, and migration of cancer cells [37]. In the presence of EIE (Figure 3), malignant markers, including SCC, UGF, interleukin-2 (IL-2) receptor, Muc-16, and vimentin, were strongly downregulated in EC cells. These results showed that the two miRNAs in the EIE strongly suppressed the migration and invasion of the peripheral EC cells.

Second, the three substances activated senescence in EC cells, and senescence was intensely triggered under AH1000 (Figure 2d). Based on miRNA profiling in the EIE, specific miRNAs, such as autophagy suppression and DNA repair, were upregulated in the presence of AH1000 (Table 1). Interestingly, anti-Unc-51-like kinase 1 (ULK1) [38], hsa-miR-423-5p was downregulated 100 times, and several anti-DNA repair genes hsa-miR-93-3p (Table 1) were upregulated 8.1 times. The exosomal results corresponded with the senescence results (Figure 2d), and AH activated cellular senescence through the activation of cytotoxic autophagy and suppression of DNA repair. Furthermore, the EIE were potent for cellular senescence acceleration through the suppression of DNA repair in EC cells. As described in this study (Table 1), hsa-miR-615-3p and hsa-let-7e-5p, which target several proteins associated with DNA repair, were upregulated in the EIE.

The MMP significantly decreased in the presence of AH500 and AH1000, and was 2.92 times lower than that in the control. The exosomal results showed that anti-ATP5A1 and ATP5I hsa-miR-877-3p were 6.7 times upregulated (Table 1). Previous studies [39,40] have reported that *ATP5A1* and *ATP5I* genes activate ATP synthesis on the electron transport system in mitochondria. Although cancerous activity is based on ATP synthesis activation, AH suppresses ATP synthesis by inhibiting mitochondrial membrane activity.

Third, miRNAs in EIE affect endocrine modulation, such as estrogen and prostaglandin synthesis and signaling. Anti-*HSD17B12* hsa-miR-874-3p was significantly downregulated, whereas anti-*STRN3* hsa-miR-125b-5p, anti-*PHB2* hsa-miR-10a-5p, and anti-*ISL1* hsa-let-7d-5p were significantly downregulated in EIE (Table 1 and Figure 4 and Figure 5). Additionally, EIE strongly attenuated the expression of estrogen and androgen receptors in ovarian cancer cells (Figure 4b). Unlike the HSD17B12 protein, estradiol synthesis enzyme [41], STRN3, PHB2, and ISL1 are associated with negative regulation of estrogenic intracellular signaling [42,43,44]. Likewise, excess estrogen causes several diseases, including diabetes mellitus and gynecological cancers, suggesting that EIE prevents carcinogenesis and cancer deterioration by the hormones. Furthermore, *PTGS2* targeting hsa-miR-1908-5p was strongly increased in EIE. The PTGS2 protein, also known as cyclooxygenase-2 (COX-2), synthesizes prostaglandins from arachidonic acid. The synthesized prostaglandin contributes to tumorigenesis, including proliferation, metastasis, and angiogenesis [45]. The significant increase in hsa-miR-1908-5p expression in EIE indicates a strong suppression of tumorigenesis by attenuating COX-2 in peripheral EC cells.

Fourth, AH and EIE were involved in attenuating drug resistance and autophagy in EC cells. In the presence of AH, *ABCB1* expression was decreased by miR-451a in the AH-induced exosomes. When exposed to an anticancer drug, cancer cells alter their phenotypes to resist the drug by increasing the ATP binding cassette subfamily B member 1 (ABCB1) protein [46]. Furthermore, ABCB1 expression is modulated by various transcription factors, including P53 [47], NF-κB [48], and YB-1 [49]. In contrast to the alteration of hsa-miR-451, hsa-miR-423-5p, hsa-miR-19b-3p, and hsa-miR-320a levels in EIE were significantly decreased, and notably, hsa-miR-423-5p expression was 100 times lower than that in the control. Interestingly, corresponding to the results for miRNA profiling (Table 1), EIE downregulated the expression of drug-resistant genes, including *NF-κB (P50, P52)*, *mTORC2*, and *ABCB1* (Figure 3). Activated *NF-κB (P50, P52)* and *mTORC2* induce multidrug resistance [50,51] and suppress autophagy [52,53]. These results show that EIE intensely enhances autophagy and attenuates drug resistance in EC cells.

Fifth, the downregulated hsa-let-7e-5p contributed to suppressing the innate immune response by upregulating the POLR3D protein, a subunit of RNA polymerase [54]. POLR3D activates IFN-β production, associated with the downregulation of inflammatory responses [55]. In agreement with a previous study [20], AH attenuated the inflammatory response through EIE from EC cells.

## 4. Materials and Methods

### 4.1. Cell Culture

Human EC (ATCC HTB115, VA, USA) and ovarian cancer (OVCAR3, ATCC HTB 161) cells were purchased and cultured in Dulbecco’s modified Eagle’s medium (DMEM; Gibco, Waltham, MA, USA) containing high glucose. After ripening at 10 °C for 48 h, *A. helianthus* flowers were exposed to infrared rays for 2 h and ground into a powder (400 mesh). The powder was extracted and concentrated under 0.08 MPa, 70 °C, and 2 h, and filtered with a 0.2 μm syringe filter (Nalgene, Waltham, MA, USA). After treatment with dosages, the extracts (100, 500, 1000 μg/mL) supported by Life Together (Gangwon-Do, Korea), 5 μg tocopherol (Sigma, St. Louis, MO, USA), and 25 µg rutin (Sigma), the endometrial cancer cells were cultured for 72 h at 37 °C in an atmosphere containing 5% CO_2_. The ovarian cancer cells were cultured with 50 μL/mL EIE for 72 h at 37 °C in an atmosphere containing 5% CO_2_. To estimate the exosomal functions, EC and ovarian cancer cells were cultured with 20 µL EIE. The total volume of medium per well was 1 mL.

### 4.2. Semi-Quantitative PCR

Total RNA was isolated from cells exposed to the three substances using RiboEx reagent (GeneAll; Seoul, Korea). cDNA was synthesized using Maxime RT PreMix (iNtRON; Seongnam, Korea), and quantitative PCR was performed with the primers (Appendix A) using the following parameters: 1 min at 95 °C, 35 cycles of 35 s at 59 °C and 1 min at 72 °C.

### 4.3. Flow Cytometry Analysis

To analyze the expression levels of markers, including CK8, Muc-16, and vimentin, the cells were fixed with 2% paraformaldehyde for 24 h, treated with 0.02 Tween 20 for 5 min, and incubated with three fluorescence-labeled three immunoglobulins, FITC anti-cytokeratin 8 (Abcam, Cambridge, UK), PE-anti-CA125 (Santa Cruz, CA, USA), and APC-anti-vimentin (Santa Cruz, CA, USA) for 48 h at room temperature with agitation at 85 rpm. The washed samples were analyzed using a flow cytometer, BD FACScalibur (BD Biosciences, San Diego, CA, USA) and FlowJo 10.6.1 (BD Biosciences, San Diego, CA, USA).

### 4.4. Migration Test

To estimate the migration inhibitory efficacy of the three substances, EC cells were cultured using the Radius^TM^ 24-well cell migration assay kit (Cell Biolabs, Inc., San Diego, CA, USA). After excluding the gel spots, the seeded cells were cultured in all areas in the wells. After detaching the gels, the migration was estimated at specific time intervals (0, 12, 24, 36, 48, and 60 h) using an ECLIPSE Ts2 instrument (Nikon, Tokyo, Japan) and the Nikon NIS Elements V5.11 (Nikon, Tokyo, Japan) imaging software.

### 4.5. Invasiveness Test

To estimate the invasion inhibitory efficacy of the three substances, EC cells were cultured in a CytoSelect^TM^ 24-well cell invasion assay kit (Cell Biolabs, Inc., San Diego, CA, USA). Invasiveness was estimated using a flow cytometer (BD FACScalibur, (BD Biosciences, San Diego, CA, USA) and FlowJo 10.6.1 (BD Biosciences, San Diego, CA, USA).

### 4.6. MMP Test

After exposure to the three substances for three days, EC cells were stained using the JC-1 Mitochondrial Membrane Potential Assay Kit (Invitrogen, Carlsbad, CA, USA) and the mitochondrial activity was estimated using a flow cytometer (BD FACScalibur, BD Biosciences, San Diego, CA, USA) and FlowJo 10.6.1 (BD Biosciences, San Diego, CA, USA).

### 4.7. Senescence Test

After exposure to the three substances for three days, EC cells were stained using the CellEvent™ Senescence Green Flow Cytometry Assay Kit (Invitrogen). The stained cells were measured using a flow cytometer (BD FACSCalibur, BD Biosciences, San Diego, CA, USA) and FlowJo 10.6.1 (BD Biosciences, San Diego, CA, USA).

### 4.8. Exosome Purification and miRNA Profiling

Semi-confluent EC cells were cultured for three days in the absence of FBS. The supernatants were filtered with 0.8 µm syringe filters (Sigma, Seoul, Korea), and the induced exosomes in the supernatants were isolated and their purity was estimated using the exoEasy Maxi kit (QIAGEN; Venlo, The Netherlands) and FITC anti-CD63 immunoglobulin (ab18235, Abcam, Cambridge, MA, USA), respectively. The FITC anti-CD63-positive exosomes were estimated using a flow cytometer, Carlibur (BD Biosciences, San Diego, CA, USA). The induced exosomes from the samples (*n* = 3) were summated for miRNA profiling in the presence of each substance. Small microRNAs were sequenced by ebiogen Inc. (Seoul, Korea) to analyze exosomal functions. An Agilent 2100 bio-analyzer and the RNA 6000 PicoChip (Agilent Technologies, Amstelveen, The Netherlands) were used to assess RNA quality. RNA was quantified using a NanoDrop 2000 spectrophotometer (Thermo Fisher Scientific, Waltham, MA, USA). The Agilent 2100 Bio-analyzer instrument for the high-sensitivity DNA assay (Agilent Technologies, Inc. Santa Clara, CA, USA) and NextSeq500 system single-end 75 sequencing (Illumina, San Diego, CA, USA) were used to prepare and sequence small RNA libraries. To obtain a bam file (alignment file), the sequences were mapped using bowtie2 software (CGE Risk, Lange Vijverberg, Netherlands), and the read counts were extracted from the alignment file using bedtools (v2.25.0) (GitHub, Inc., San Francisco, CA, USA) and R language (version 3.2.2) (R studio, Boston, MA, USA) to determine the miRNA expression level. miRWalk 2.0 (Ruprecht-Karls-Universität Heidelberg, Medizinische Fakultät Mannheim, Germany) was used for miRNA target signal study, and ExDEGA v.2.0 (ebiogen Inc., Seoul, Korea) was used to deduce various results, including van diagram, heatmap, scattering plots, and bar graphs.

### 4.9. Statistical Analysis

The data were analyzed by one-way analysis of variance (ANOVA) with the post hoc test (Scheffe’s method) and independent t test using the SPSS software v26 (IBM, New York, NY, USA) and Prism 7 software (GraphPad Software Inc., San Diego, CA, USA).

## 5. Conclusions

In summary, AH significantly modulated the tumorigenic microenvironment in EC cells. Although α-tocopherol and rutin contents in *A. helianthus* were high, AH modulated the environment more effectively than the two substances. The modulations by AH were summarized in five biological categories: migration and invasion suppression, increased drug susceptibility, decreased inflammation, reproductive cancer activity attenuation, and cellular senescence activation. Furthermore, the effects of EIE suggest a potency to protect against and heal gynecological cancers. These functional miRNA candidates should be further studied via in vitro and in vivo experiments with liposomes containing functional miRNAs. These candidates will help in performing in vivo and in vitro experiments to estimate the potency of a drug for gynecological cancers.

## Figures and Tables

**Figure 1 molecules-26-02207-f001:**
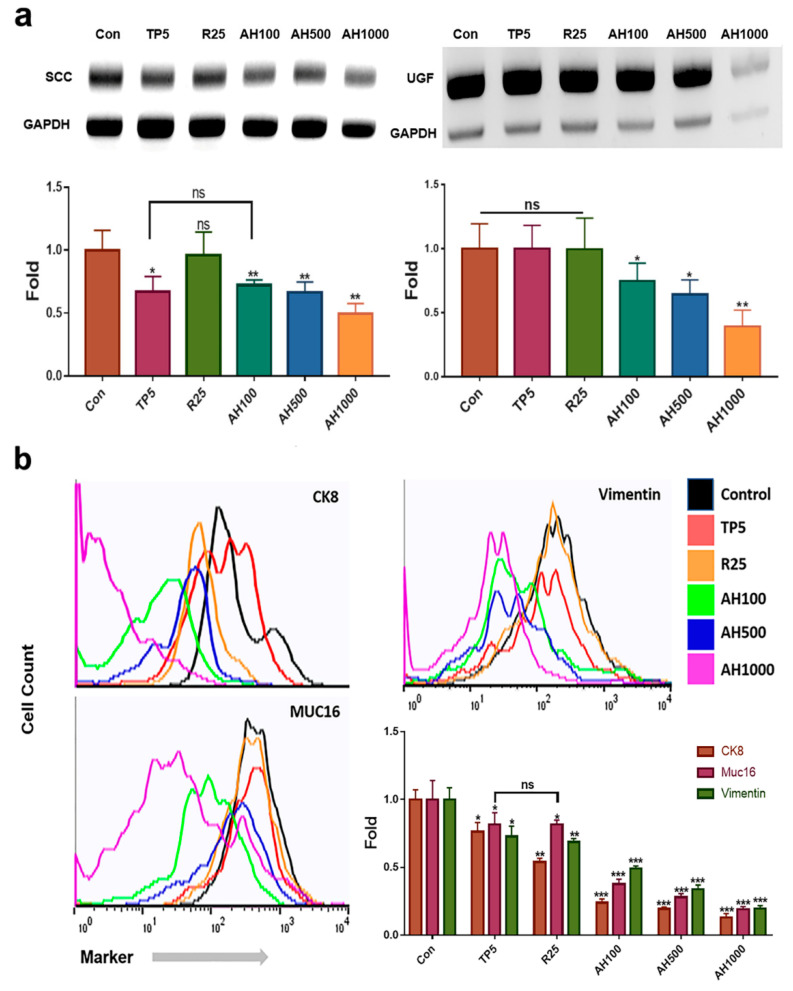
Levels of endometrial cancer (EC) cell markers in the presence of bioactive substances. (**a**) qPCR results indicate that the expression levels of markers of the endometrial cancer cells, such as squamous cell carcinoma antigen (SCC) and urinary gonadotropin fragment (UGF), exposed to bioactive substances, including 5 μg/mL tocopherol-α (TP5), 25 μg/mL rutin (R25), and 100, 500, and 1000 μg/mL *A. helianthus* extract (AH100, AH500, and AH1000, respectively), were downregulated. (**b**) CK8^+^, Muc16^+^ and Vimentin^+^ cells were counted using a flow cytometer. ns: not significant; * *p* < 0.05; ** *p* < 0.01; *** *p* < 0.001.

**Figure 2 molecules-26-02207-f002:**
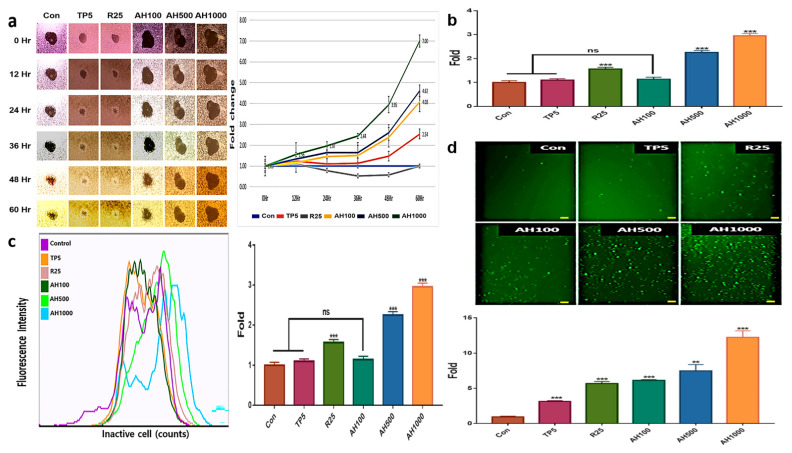
Inhibition of tumor activity by bioactive substances in endometrial cancer (EC) cells; (**a**) The central areas indicate the unseeded spaces with EC cells. The sizes were inversely proportional to EC cell migration. The areas were measured using Nikon software and the measured values indicate the relative fold changes. (**b**) The bar graphs indicate the non-invasive cells in the presence of 5 μg/mL tocopherol-α (TP5), 25 μg/mL rutin (R25), and 100, 500, and 1000 μg/mL *A. helianthus* extract (AH100, AH500, and AH1000, respectively). (**c**) The specific fluorescence indicates inactive cells using mitochondrial membrane potential. (**d**) The senescenced cells were stained with green fluorescent dye, and the stained cells were counted using a flow cytometer, (scale bars = 100 μm). ns: not significant; * *p* < 0.05; ** *p* < 0.01; *** *p* < 0.001.

**Figure 3 molecules-26-02207-f003:**
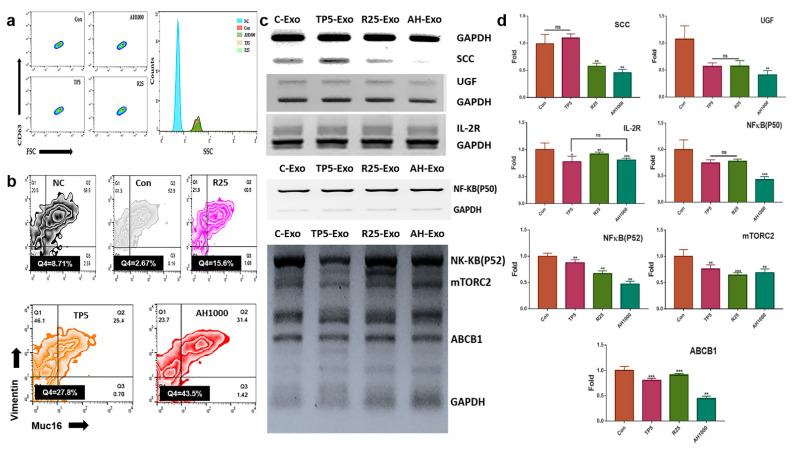
Expression of cancerous markers in endometrial cancer (EC) cells in the presence of induced exosomes; (**a**) The histograms indicate the induced exosomes from EC cells. NC was not stained with FITC anti-CD63. (**b**,**c**) The gated population; CD63^+^ exosomes were analyzed in the presence of FSC and SSC. The graphs indicate the expression of the markers in EC cells in the presence of exosomes. (**d**) Muc16^+^ and vimentin^+^ cell count in EC cells in the presence of exosomes; 5 μg/mL tocopherol-α (TP5), 25 μg/mL rutin (R25), and 100, 500, and 1000 μg/mL *A. helianthus* extract (AH100, AH500, and AH1000, respectively). ns: not significant; * *p* < 0.05; ** *p* < 0.01; *** *p* < 0.001.

**Figure 4 molecules-26-02207-f004:**
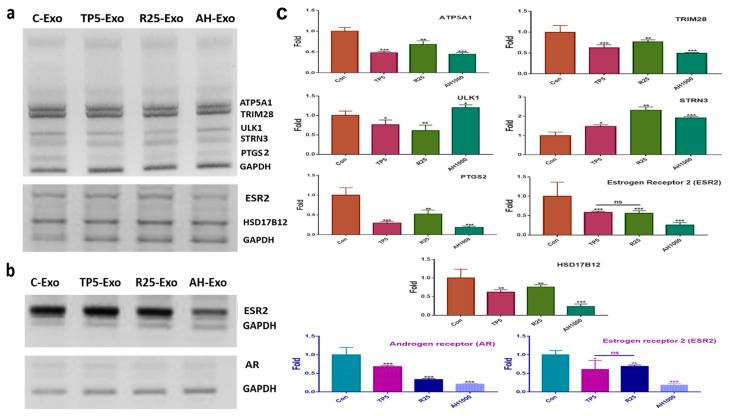
Effects of induced exosomes on various parameters associated with endometrial cancer (EC) cells; (**a**) Effects of miRNAs on biochemical marker levels. (**b**) Effects of miRNAs on ovarian cancer cells; 5 μg/mL tocopherol-α (TP5), 25 μg/mL rutin (R25), and 1000 μg/mL *A. helianthus* extract (AH1000). (**c**) Histograms for the a and b panels. C-Exo, exosomes from EC cells; TP5-Exo, exosomes from TP5-treated EC cells; R25-Exo, exosomes from R25-treated EC cells; AH-Exo, exosomes from AH1000-treated EC cells. ns: not significant; * *p* < 0.05; ** *p* < 0.01; *** *p* < 0.001.

**Figure 5 molecules-26-02207-f005:**
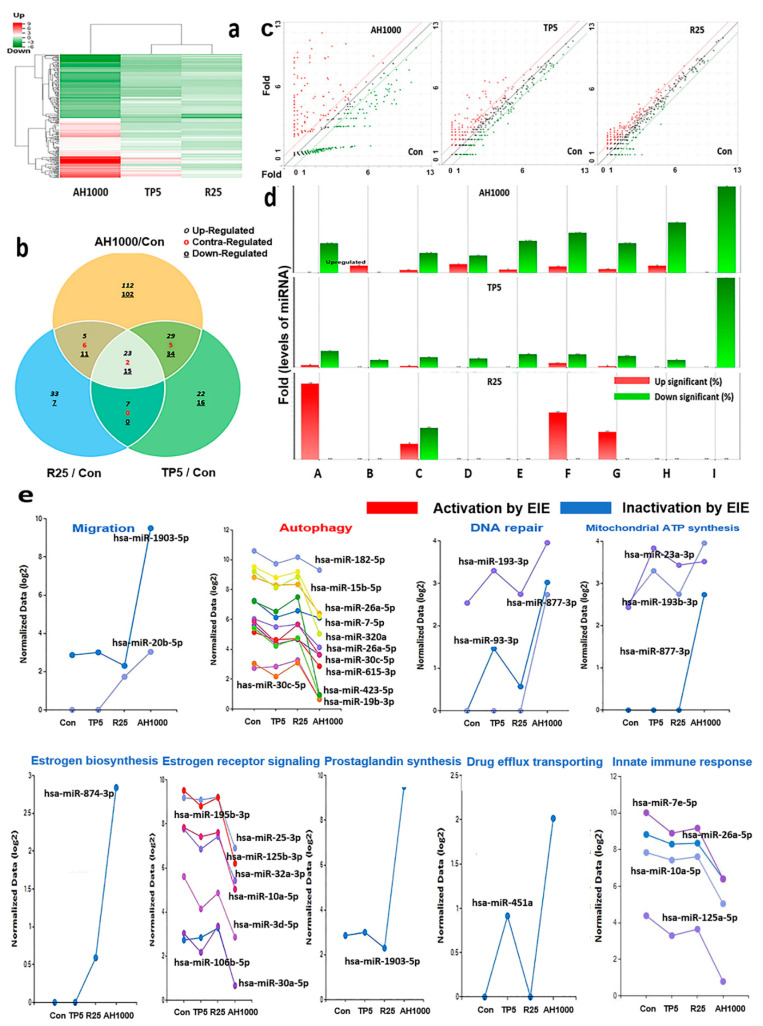
Profiling of miRNAs in the induced exosomes from endometrial cancer (EC) cells; (**a**) The heatmap for microRNA in exosomes induced by the three substances. (**b**) The Venn diagram for microRNAs in EC cells in the presence of three substances. (**c**) The scattering plots of the control and bioactive substances. (**d**) The microRNAs involved in several categories (A: positive regulation of autophagy, B: drug transmembrane transport, C: positive regulation of cell migration involved in sprouting angiogenesis, D: estrogen biosynthetic process, E: positive regulation of prostaglandin biosynthetic process, F: mitochondrial ATP synthesis coupled proton transport, G: positive regulation of DNA repair and regulation of innate immune response, H: negative regulation of intracellular estrogen receptor signaling pathway, I: positive regulation of DNA repair. (**e**) The biological classification of significant microRNAs in the induced exosomes; 5 μg/mL tocopherol-α (TP5), 25 μg/mL rutin (R25), and 1000 μg/mL *A. helianthus* extract (AH1000). 2 times ≤ fold changes and 0.5 times ≥ fold changes; * *p* < 0.05; ** *p* < 0.01; *** *p* < 0.001.

**Table 1 molecules-26-02207-t001:** Significant miRNAs involved in the five target biochemical categories and their modulating genes.

Table		miRNA Symbol	Fold	Evaluating Gene *
**I**	Positive regulation of cell migration involved in sprouting angiogenesis	hsa-miR-1908-5p	100	*PTGS2*
		hsa-miR-20b-5p	8.2	*VEGFA*
**II**	Positive regulation of autophagy	hsa-miR-423-5p	100	*ULK1*
		hsa-miR-19b-3p	25	*PRKAA1*
		hsa-miR-320a	20	*ULK1*
	Positive regulation of DNA repair	hsa-miR-93-3p	8.1	*NPM1, OTUB1, PARP1, SMC1A, TRIM28*
		hsa-miR-877-3p	6.7	*SUPT16H, TRIM28, HMGB1*
	Mitochondrial ATP synthesis coupled proton transport	hsa-miR-877-3p	6.7	*ATP5A1, ATP5I*
**III**	Estrogen biosynthetic process	hsa-miR-874-3p	7.2	*HSD17B12*
	Negative regulation of intracellular estrogen receptor signaling pathway	hsa-miR-125b-5p	10	*STRN3*
		hsa-miR-10a-5p	6.9	*PHB2*
		hsa-let-7d-5p	6.8	*ISL1*
	Positive regulation of prostaglandin biosynthetic process	hsa-miR-1908-5p	100	*PTGS2*
**IV**	Drug transmembrane transport	hsa-miR-451a	4	*ABCB1*
**V**	Regulation of innate immune response	hsa-let-7e-5p	12.5	*POLR3D*
**Up regulation**		**Down regulation**		

* Gene expression is modulated by the alteration of miRNA target genes in the five categories.

## Data Availability

Not applicable.

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
