# Peer review of "Effects of Induced Exosomes from Endometrial Cancer Cells on Tumor Activity in the Presence of Aurea helianthus Extract"

_molecules, 2021, doi:10.3390/molecules26082207_

Round 1

Reviewer 1 Report

In the manuscript entitled, “Effects of Induced Exosomes from endometrial cancer cells under Aurea helianthus Extract on Tumor Microenvironment” the authors determined the protective effects and underlying mechanisms of A. helianthus plant extract using endometrial cancer (EC) as experimental model. Overall, the findings demonstrated that A. helianthus-induced release of bioactive exosomes-containing miRNAs (EIE) resulted in the suppression of migration and invasion of EC cells. Importantly, these effects were found to be mediated via mechanisms, including activation of cellular senescence leading to reduced mitochondrial membrane potential and increased autophagy/drug susceptibility, as well as attenuation of reproductive cancer activity and inflammation. However, there are several concerns that need to be addressed.  

Major comments:

  1. Figure 2 is confusing. The title states that “Inhibition of tumor activity by bioactive substances in EC cells. However, in Fig 2a, “the analysis of the areas without migrated cells” does not make any sense and does not proof the anti-tumor activity of AH compound as compared to TP5 and R25 compounds. Again, Fig 2b indicates that AH compound (at 1000µg/ml concentration) is increasing cell invasiveness, which is opposite to what is being tested. Only, Fig 2d indicated that AH treatment induces senescence. Authors should analyze and present the data differently to reflect this effect. The significance needs to be included (e.g., *, **, ***) for each of these figure panels.
  2. In figure 1, the non-significant effect differences between various treatments groups are indicated. However, the significance between other treatment groups, e.g., dose-dependent effects of AH is not indicated, while p<0.05 is mentioned in the legend. This should be fixed.
  3. The significance needs to be included (e.g., *, **, ***) for data presented in Folds in figure 3 panels. The same goes for other figures.
  4. MiRNAs profiling with their suggestive functions is shown in figure 5. However, it is not clear if the functions of these miRNAs are verified by functional analysis (e.g., inhibition of cell migration by has-miR-20b-5p).

Author Response

I appreciate for your comments to improve our manuscript

Reviewer 2 Report

Comments on the manuscript (ID: molecules-1143080) “Effects of Induced Exosomes from endometrial cancer cells under Aurea helianthus Extract on Tumor Microenvironment” by Park Yoonjin et al. to be published in Molecules (MDPI).

This manuscript deals with the effects of the A. Helianthus effects in vitro of exosome induction in endometrial cancer cells. The manuscript needs to be fully revisited in terms of materials and methods (poorly described), English grammar (to be carefully checked), results (not sufficiently described), experiments and conclusions (not sufficiently supported by results). This is enlisted below in major and minor compulsory points. The Authors are suggested to resubmit a completely revisited form of this manuscript, with particular attention to the English grammar and items/figures description.

Major points:

  1. The use of the term “Tumor microenvironment” (TME) in the article’s title is inappropriate. This definition implies to mention immune cells, never experimentally investigated in this manuscript. Moreover, TME is a definition usually employed in in vivo experiments. In this regard, Authors are recommended to consider to plan in vivo expts where EC-bearing mice will be administered with the AH extract. This will enable Authors to check extract toxicity as well as production and content of EC-derived exosomes ex vivo (compared to saline-treated EC-bearing mice). In addition this will also allow Authors to compare/confirm exosome results with those obtained by figures 4-5. Otherwise, Authors must delete the term TME from title and insert the term “in vitro” to underline that the experiments are of in vitro origin.
  1. To me it is unclear why Authors only display the not significant stats and omitted to show the significant ones. The statistics are not fully clear in this context. Please depict the significant comparison too, and explain why it is important to indicate the NS comparisons (i.e., the TPS vs AH100 or Con vs R25 in figure 1A).
  1. The statements at rows 95-101 needs to be fully revisited and written in a more clear/extended manner. Indeed, the message herein depicted is almost unclear and difficult to undertsand.
  1. The Authors must to extend the size of each subpanel in the figure 1A, with specific details on the concept of the migration test in the results section, that needs to be clarified. Are the irregular dots at the center of the subpanels tumor spheroids? Please explain in an extensive and clear manner. How the Authors see the migration extent? How did the Authors obtain the data depicted in the curves adjacent to the cell subpanels? Please detail in a clear manner.
  1. Figure 2, panel D: what is the concept of the assay? This is insufficiently described in results section. How is the senescence determined? Please provide more details.
  1. Authors omitted to describe the Western blot method they employed in this manuscript. To me it is unclear why the GAPDH bands are present in the same blot with the parallel Ab specific sample bands. For instance, in figure 3 SSC signal is detected together with the GAPDH one in the same blot. Authors need to extensively clarify this issue in all the figure with WB results ( (Figs 1, 3 and 4).
  1. The discussion is an extended version of the results, and needs to be completely rewritten with emphasis on then novelty of these results and future directions. The proposed in vivo experiment may improve the impact of this manuscript by allowing Authors to extend the discussion on the possible miRNA roles found in table 1 to the infiltration of immune cells in the EC-bearing mice. For example, has-les-7e-5p induces regulation of immune responses (as depicted in table 1). This can be further explained/discussed with the results coming from such an in vivo experiment. In addition, Author can consider to examine the exosome content in the blood of EC-bearing mice treated with AH extract vs saline mice and compare these data with that displayed by figures 4-5 and table 1.

Minor points

  1. Please insert scalebars in each subpanel of panel A (figure 2).
  1. The last part of the Introduction (rows 68-75) must rewritten and extended concerning the importance and novelty of the results herein displayed.
  1. All figure legends are not sufficiently detailed and do not specify the material illustrated in the figure. Please rewrite in a more extended and clear manner.
  1. Figure 1B: control is grey in legend and black in graph.
  1. Please uniform all colour codes in all legends and graphs in all figures. Use one unique color for one unique condition throughout the manuscript whenever possible.
  1. Please insert panel labels in figure 3.

Author Response

(The authors gave the same response as above.)

Reviewer 3 Report

It is an interesting study showing that Aurea helianthus extract is able to reduce typical markers of tumor progression (invasion, migration, metabolic changes) in one cell line of endometrial cancer. This study also shows that endometrial cancer cells exposed to the above-mentioned extract produce exosomes that have distinct miRNA profiles compared to control cells. The authors further showed that the treatment of endometrial and ovarian cancer cells with those exosomes exhibit reduced mRNA and protein expression levels of different tumor markers compared to controls.

Minor comments:

- Figure 1a: it is clear that extract is able to modify GAPDH levels. Try other protein loading marker (i.e. actin, tubulin) and quantify SCC and UGF again.

- Figure 1a: place the image of the protein loading marker below SCC.

- Statistical analysis is completely missing in the whole manuscript. Please, show asterisks or another symbol to indicate any statistical difference among experimental groups.

- Figure 2a: graph has a poor image quality. Improve it, please.

- Figure 2d: provide a better microscope image.

- Show gating strategy to isolate CD63+ exosomes.

- Include a detailed description on how exosomes where isolated from cell cultures.

- Include a detailed description on how cells were treated with the exosomes isolated from cell culture. How many exosomes were added to the cell culture for treatments? Clearer methods must be shown in the manuscript.

- Figure 3. Panel letters (a, b,…) are missing. Please, include them.

Author Response

(The authors gave the same response as above.)

Round 2

Reviewer 1 Report

The authors have satisfactorily addressed the comments and incorporated the suggestions in the revised manuscript. I have no further comments.

Author Response

Thank you very much for your comments. 

Reviewer 2 Report

The Authors need to clearly trace each point in their response to this reviewer's questions. In this way the yellow changes can be associated to each precise Author's response to the reviewer's comments. Please insert the exaxt row intervals in each response and clearly describe/justify/explain the responses, paying particular attention to those requiring experimental implementations (see below).

The Authors need to provide the in vivo experiment with EC-beraring mice (see major comments no. 1 and 7).

The PCRs showed in figure and in M&M are NOT quantitiative but semi-quantitative. Indeed the quantitation histograms are made from the image quantitation of bands, and NOT from a delta-Ct threshold value provided in case of true quantitative PCRs.

Overall, this revised version is still not sufficiently implemented/revised to be accepted for publication in Molecules.

Author Response

The Authors need to provide the in vivo experiment with EC-beraring mice (see major comments no. 1 and 7). 

Answer1. We described the EC-bearing in mice associated with our results for EIE in discussion

The PCRs showed in figure and in M&M are NOT quantitiative but semi-quantitative. Indeed the quantitation histograms are made from the image quantitation of bands, and NOT from a delta-Ct threshold value provided in case of true quantitative PCRs.

Answer2. We revised the subheading.  

Overall, this revised version is still not sufficiently implemented/revised to be accepted for publication in Molecules. 

Answer3. Overall, we edited sentences and english style in our manuscript.